# “INTEGRO INTEGRated Psychotherapeutic InterventiOn” on the Management of Chronic Pain in Patients with Fibromyalgia: The Role of the Therapeutic Relationship

**DOI:** 10.3390/ijerph20053973

**Published:** 2023-02-23

**Authors:** Ilenia Pasini, Cinzia Perlini, Valeria Donisi, Anna Mason, Vittorio Schweiger, Erica Secchettin, Fabio Lugoboni, Gaetano Valenza, Lidia Del Piccolo

**Affiliations:** 1Department of Neurosciences, Biomedicine and Movement Sciences, University of Verona, 37134 Verona, Italy; 2Pain Therapy Centre, Department of Surgery, Dentistry, Maternal and Infant Sciences, Verona University Hospital, Policlinico GB Rossi, 37134 Verona, Italy; 3Unit of Addiction Medicine, Department of Internal Medicine, Integrated University Hospital of Verona, Policlinico GB Rossi, 37134 Verona, Italy; 4Bioengineering and Robotics Research Center “E. Piaggio”, Department of Information Engineering, School of Engineering, University of Pisa, 56122 Pisa, Italy

**Keywords:** fibromyalgia, integrated psychotherapy, attunement, psychophysiological synchronization, chronic pain, therapeutic alliance, cognitive behavioral therapy (CBT), psychophysiology

## Abstract

Fibromyalgia (FM) is a chronic disease characterized by a heterogeneous set of physical and psychological conditions. The chronic experience of disability felt by patients and the impact on quality of life (QoL) of the disease may worsen the cognitive reappraisal ability and contribute to maintaining an altered pain modulation mechanism. This paper presents the study protocol of an INTEGRated psychotherapeutic interventiOn on the management of chronic pain in patients with fibromyalgia (INTEGRO). The aim of the study is to investigate the efficacy of an integrated psychotherapeutic intervention focused on pain management on QoL and pain perception, in a pilot sample of 45 FM patients with idiopathic chronic pain. The contribution of perceived therapeutic relationship (alliance) and physiological attunement, in both the patient and therapist, will be considered as possible mediators of intervention efficacy. Attachment dimensions, traumatic experiences, difficulties in emotion regulation, mindfulness attitude and psychophysiological profile will also be considered as covariates. The objectives are to evaluate longitudinally if patients will experience an increase in QoL perception (primary endpoint), pain-managing self-efficacy and emotion-regulation abilities as well as a reduction in pain intensity (secondary endpoints), considering the mediating role of perceived therapeutic alliance and physiological attunement in both the patient and therapist.

## 1. Introduction

Fibromyalgia (FM) is a chronic disease with a world prevalence of around 2–6%, according to different epidemiological reports [1,2], and 2.2% in Italy [3]. Patients with FM report mainly chronic and diffuse musculoskeletal pain [4], together with a heterogeneous set of other conditions, such as physical and mental fatigue, disturbed sleep, headaches, cognitive deterioration and other functional disorders [5].

The etiology of FM is not entirely clear, although several studies describe FM in terms of a central sensitization syndrome, characterized by the alteration of nociceptive process of the central nervous system, distorted pain modulation mechanisms and amplification of pain experience due to stress and psychological factors [6,7]. For instance, Montoya et al. [8] highlighted an increase in perceived pain when FM patients viewed unpleasant images, which was different from patients with musculoskeletal pain due to organic lesions, suggesting that these patients are particularly sensitive to unpleasant emotional stimuli.

Moreover, FM patients show a psychological profile characterized by high levels of anxiety, depression, high focus on algic stimuli, catastrophizing and greater sensitivity in the dynamics of interpersonal identification and social comparison [8,9]. A recent observational study, conducted by our group, showed that in FM patients, there is a correlation between a history of depressive symptoms and poor quality of life (QoL) [10]. FM patients often also report feeling rejected, ignored and “not taken seriously” [11,12].

FM syndrome requires not only a specialist assessment of pain, disability and co-morbidity, but also a psychosocial assessment. Therefore, the management of FMs should be based on a multimodal method (pharmacological treatment, physical activity and psychotherapeutic support) and should aim to improve health-related quality of life [13,14,15], also considering the possible role of drug interactions and related risks of drug abuse [16] due to the type of pharmacological treatment suggested by clinical guidelines [14]. FM pharmacological options include: analgesic (i.e., opioids and paracetamol), anticonvulsant (i.e., pregabalin), antidepressant (i.e., fluoxetine and paroxetine, duloxetine and amitriptyline), muscle relaxant (i.e., cyclobenzaprine) and cannabinoid drugs.

### 1.1. Psychotherapy in Fibromyalgia Syndrome

Chronic pain is one of the most relevant symptoms in FM. The literature shows that psychotherapy may be particularly useful for the treatment of chronic pain whose etiology is strongly related to psychological factors [13,17,18]. Indeed, the experience of pain and the resulting disability are largely moderated either by negative and irrational beliefs, compensatory behaviors or uncontrolled physiology. Chronic activation of stress circuits is recognized as one of the key factors in pain [19]. In fact, many FM patients achieve a diagnosis only after several specialist visits, which contributes to increased stress perception, enhancing the risk for the onset of depressive and anxious symptoms. Therefore, the cognitive reappraisal of stressful events may allow a containment of both emotional and somatic responses, contributing to improving psychobiological allostasis (the process by which a state of internal physiological equilibrium is maintained in response to actual or perceived environmental and psychological stressors) [20].

Cognitive behavioral therapy (CBT) has been shown to significantly contribute to reducing painful symptoms in patients with chronic pain [21,22,23,24,25,26,27]. The primary outcomes after CBT treatment were: pain relief (improvement ≥ 50%), improvement of health-related quality of life (≥20% improvement of HRQoL) [28], reduction in thymic deflection and improvement of the perception of self-efficacy in pain management [29].

Moreover, acceptance and commitment therapy—ACT (CBT third generation intervention)—has been shown to improve the acceptance of pain and the reduce pain catastrophizing, compared to drug therapy only [30]. Most studies report group CBT interventions, either as a single treatment or as part of a so-called multicomponent or multidisciplinary therapy program (where CBT is combined with any other defined active therapy, such as physical exercise, physical therapy or drug therapy with defined extent and intensity) [31,32]. Among these, Thieme, Flor and Turk [33], Luciano et al. [34] and Pérez-Aranda et al. [35] demonstrated the usefulness of active and integrated psychological interventions (such as psychoeducation, behavioral techniques, mindfulness, analysis of dysfunctional thoughts and cognitive restructuring strategies) compared to usual treatment only.

### 1.2. Interpersonal Psychophysiology and Therapeutic Alliance

A rather recent line of research is to study the relationship between the degree of interpersonal attunement and psychophysiological synchronization, expressed as the correlation between continuous measurements of autonomic nervous system activity in two or more people who interact with each other [36]. Psychophysiological synchronization is considered as a proxy of affective attunement [37,38,39,40,41], highlighting subtle changes in the arousal and affective state that provide clues to the quality of interpersonal relationship and the effectiveness of communication styles and behaviors [42,43].

The role of psychophysiological synchronization was underlined by Marci et al. [41], who showed that the overall correlation index of skin conductance during psychotherapy sessions was positively and significantly correlated to an empathic perception of the therapist by the patient. In the phases of high skin conductance concordance, the socio-emotional interactions were significantly more positive for both patients and therapists, compared to moments of low concordance [41]. Based on these observations, it seems possible that there are different forms of empathy involved in the therapeutic relationship and that the physiological concordance index reflects a form of sensory empathy based on somatic resonance.

More generally, therapeutic alliance is a deep empathy state, connected to limbic area activity, that allows awareness of others’ emotional states [44]. The capability to create and maintain a therapeutic bond reflects, at least in part, the co-regulation and synchronization mechanism learned in the early mother–child dynamics [45,46], so much so that the presence of a secure attachment style in the patient seems to show a significant effect in the delay dynamics of physiological synchronization within the clinician–patient dyad [36,47].

Physiological synchronization in psychotherapy has mainly focused on autonomic nervous system measures, specifically cardiac (ECG) and electrodermal activity (EDA). EDA and ECG are good indicators of sympathetic and parasympathetic nervous system activity and both contribute to describing self-regulation processes and emotional responses. They also contribute to measuring unaware processes that cannot be consciously controlled, or that cannot be evaluated directly by the clinician [40,48]. However, the role of physiological processes in self-regulation and co-regulation need further investigation when considering patients who may be more sensitive to the responses given by doctors to their emotional states and their concerns, as FM patients may be.

As regards this topic, Thieme et al. [49] demonstrated that patients with FM show lower HR (heart rate) and high reactivity of SCL (skin conductance level) in reaction to stressful situations compared to individuals without pain, suggesting unique psychophysiological characteristics related to stress reaction in these patients [50,51].

Similarly, Finset et al. [52] showed that FM patients had higher levels of electrodermic activity (EDA) in an experimental situation associated with an empathic communication style, confirming that these patients perceive affective processing as a stressful condition [52]. Usually, empathic and patient-centered communication is expected to attenuate the level of self-autonomic arousal, contributing to stress reduction [42,53,54], but this was not the case for FM patients.

Given the reported evidence of CBT and ACT interventions and the role of interpersonal psychophysiology in therapeutic alliance, we designed a pilot study based on an INTEGRated psychotherapeutic interventiOn devoted to the management of chronic pain in patients with fibromyalgia, named INTEGRO.

The main study objective is to evaluate if patients will experience an increase in QoL perception after INTEGRO treatment.

The secondary objectives are to verify longitudinally if:there is a reduction in pain intensity;there is an increase in perceived self-efficacy in managing pain;there is an increase in patients’ emotional regulation ability.

All these objectives will be longitudinally measured by comparing pre-treatment (T0) to post-treatment (T13) evaluations, considering the mediating role of perceived therapeutic relationship (alliance) and physiological attunement, in both the patient and therapist.

## 2. Materials and Methods

INTEGRO is an exploratory longitudinal prospective study promoted by the Clinical Psychology Unit in collaboration with the Pain Therapy Unit of the Integrated University Hospital (Azienda Ospedaliera Universitaria Integrata—AOUI) of Verona, Italy.

The Unit of Clinical Psychology will be responsible for the research project management and the psychotherapy intervention. CBT psychotherapists, also trained in the use of relaxion and mindfulness-based techniques, later named as “therapists” will manage the psychotherapy-integrated intervention, together with the pre and post evaluations. The medical staff of the Pain Therapy Unit will be responsible for the recruitment of patients who will satisfy the inclusion criteria provided by the protocol.

### 2.1. Study Population and Recruitment

45 patients with FM diagnosis according to established ACR criteria (American College of Rheumatology [55]) and idiopathic chronic pain will be recruited by the Pain Therapy Unit. Other eligible criteria are 18–65 years old, Italian-speaking and able to provide informed consent.

Exclusion criteria are:heart disease (i.e., cardiopulmonary disease, cardiomyopathy, clinically significant arrhythmias or the presence of electronic devices);endocrinological disease;being currently pregnant or lactating;medical condition capable of altering psychophysiological recording (i.e., sclerodermia, Raynaud’s syndrome, Sjogren’s syndrome, epilepsy, neurological disease, rheumatoid arthritis or other chronic inflammatory diseases with autoimmune origin, current neoplasia, obesity (BMI > 30), drug and alcohol abuse in the last 3–6 months);psychosis and major depressive disorder;cognitive deterioration, intellectual disability;pharmacological therapy incompatible with physiological recording;currently receiving psychotherapy.

Even the psychotherapists must not have medical conditions that could alter the recording of psychophysiological signals.

### 2.2. Procedure and Measures

As a general overview of the study, in Figure 1, we report a detailed flowchart of the process of data collection.

SCL-90 (Symptom Checklist-90-R), FIQ-R (Revised Fibromyalgia Impact Questionnaire), MPQ (McGill Pain Questionnaire), BPI (Brief Pain Inventory), DERS (Difficulties in Emotion Regulation Scale), PSEQ (Pain Self-Efficacy Questionnaire), ASQ (Attachment Style Questionnaire), TEC (Traumatic Experience Checklist), FFMQ (Five Facet Mindfulness Questionnaire), VAS-E (Visual Analogue Scale-Empathy), SEP (Perceived Empathy Scale), WAI (Working Alliance Inventory). 

All eligible patients under care of the Pain Therapy Unit who agreed to participate to the study will undergo a first set of evaluations to define if they are suitable for the study **(T0a)**:

***Socio-demographic data sheet*** collecting gender, age, education, marital, employment and health status;

***Sheet for Clinical Data Collection-CRF*** (*Clinical Record Form*) completed by the specialist in analgesic therapy (current and remote medical history, substance use and drug therapy) and by a clinical psychologist (current and remote psychological history, psycho-social aspects and lifestyle), who, based both on psychiatric history and current clinical conditions, will deepen psychopathology evaluation using the ***Structured Clinical Interview-Clinician Version-SCID-CV*** [56,57] as a tool to confirm exclusion criteria when needed, similar to Botella et al. [58].

***Symptom Checklist-90-R*** (*SCL-90* [59,60], Italian version [61]). This is a self-report questionnaire that evaluates the severity of a broad spectrum of psychopathological symptoms, distinguished as internalizing (depression, somatization, anxiety) or externalizing (aggression, hostility, impulsivity) over the past seven days. Each item is rated on a 5-point Likert scale ranging from ‘Not at all’ (0) to ‘Extremely’ (4). The checklist consists of nine subscales: somatization (SOM), obsessive compulsive (O-C), interpersonal sensitivity (I-S), depression (DEP), anxiety (ANX), hostility (HOS), phobic anxiety (PHOB), paranoid ideation (PAR) and psychoticism (PSY). Different possible indexes may be calculated as the final score: positive symptom total (PST), positive symptom distress index (PSDI) and global severity index (GSI).

We chose not to apply a standardized but time-consuming clinical evaluation, such as SCID, to all FM patients. This because they already had several tests to complete. We preferred to include a quicker screening test for clinical symptoms such as SCL-90. This seems to be a good compromise between evaluating the presence of psychopathology for each patient and ensuring the feasibility of the study in terms of time assessment.

A second pre-treatment psychophysiological and psychological evaluation will then be conducted on selected eligible patients at the Clinical Psychology Unit **(T0b)**, by collecting the following measures:

***Italian version of the Revised Fibromyalgia Impact Questionnaire*** (*FIQ-R* [62], Italian validation and translation [63]). It is a self-report questionnaire composed of 21 items, considering “physical function” (9 items), “global impact” (2 items), and “symptom severity” (10 items), over the past 7 days. Each item is rated on a 11-point Likert scale ranging from ‘0′ to ’10′. Total scores range from 0 to 100. Cut-off values are distinguished in five levels of FM disease severity: 0–23 remission, 24–40 mild disease, 41–63 moderate disease, 64–82 severe disease and >83 very severe disease [64].

***McGill Pain Questionnaire*** (*MPQ* [65], Italian version [66]). It evaluates pain as a multifactorial experience (cognitive-evaluative, motivational-affective, sensory-discriminative). Each dimension is evaluated through 78 pain descriptive items classified into 20 subscales, (each containing 2–6 words that fall into 4 main subscales: sensory (subscales 1–10), affective (subscales 11–15), evaluative (subscales 16), and miscellaneous (subscales 17–20)). Individual words are counted, and a score is given based on position or order of rank in the set of words. The final score is the PRIr (pain rating index rank) which ranges from 0 to 78. In addition, there is a numeric–verbal combination that indicates overall pain intensity rated on a 6-point Likert scale ranging from ‘none’ (0) to ’atrocious’ (5) defined as present pain intensity (PPI) score. A higher score on the MPQ indicates worse pain, and the PRIr may be interpreted either as quantitative (number and ranking of each selected word) or qualitative (type of words chosen). The MPQ may be used as a therapeutic work tool to improve pain awareness (quality, intensity, location).

***Brief Pain Inventory*** (*BPI* [67,68], Italian validation [69,70]). It is a self-report questionnaire evaluating the intensity and interference of pain in the last 24 h. It has a three-factor structure: pain intensity, pain interference in the emotional sphere and interference in work activities. Each dimension ranges from 0 to 10. A cut-score of 5 indicates that the pain starts to interfere heavily in daily activities and has an adverse impact on quality of life.

***Pain Self-Efficacy Questionnaire*** (*PSEQ* [71], validation and Italian translation [72]). It evaluates patients’ perception and beliefs of self-efficacy in chronic pain management. It consists of 10 items reporting different daily activities (e.g., “I can do most of the household chores”) or general aspects of life (e.g., “I can still achieve most of my goals in life”), for which the patient indicates how safe they feel to carry out these activities, despite the presence of pain. Each item is rated on a 7-point Likert scale ranging from 0 (not at all safe) to 6 (completely safe). The total score of the questionnaire is the sum of items and can range from 0 to 60, with higher scores indicating greater self-efficacy in pain management.

***Attachment Style Questionnaire*** (*ASQ*, Feeney, Noller, and Hanrahan in [73], Italian translation [74]). The ASQ is composed of 40 items on a 6-point Likert scale (i.e., from 1 (totally disagree) to 6 (totally agree)) and considers five dimensions of attachment: (i) discomfort with intimacy (ASQ-DC); (ii) need for approval (ASQ-NA); (iii) concern for relationships (ASQ-PR); (iv) consideration of relationships as secondary (compared to personal success) (ASQ-SR); (v) trust in oneself and others (ASQ-C). Each subscale score will be calculated by summing up the scores for each valid case (i.e., any subscale with less than two missing items).

***Traumatic Experience Checklist*** (*TEC* [75], Italian translation by Schimmenti e Mulè). It is a self-report questionnaire inquiring about 29 types of potential trauma, including criterion A events of PTSD (“the person experienced, witnessed, or was confronted with an event or events that involved actual or threatened death or serious injury, or a threat to the physical integrity of self or others”), as well as other potential overwhelming events: loss of significant others; life threat by disease or assault; war experience; emotional neglect, emotional abuse, physical abuse, sexual harassment, sexual trauma and any other events felt traumatic by patients. For each item, it is possible to evaluate the impact that the event had on the patient (using a 5-point scale). The TEC total score ranges from 0 to 29. It is possible to calculate trauma area severity scores using four variables: (a) presence of the event; (b) age at onset, indicating whether trauma had occurred, or started, in the first 6 years of life or thereafter; (c) duration of the trauma, indicating whether trauma had lasted less or more than 1 year; and (d) subjective response, indicating whether the subject felt not traumatized or only slightly traumatized, versus moderately, severely, or extremely traumatized by the event(s). These variables are given a score of 1 if they apply, and a score of 0 if they do not.

***Difficulties in Emotion Regulation Scale*** (*DERS* [76], Italian validation [77]). It is a 36-item self-report scale asking on a 5-point Likert scale (from 1 ‘almost never’ to 5 ‘almost always’) how respondents relate to their emotions according to the following dimensions: lack of control, lack of trust, difficulty in distraction, lack of acceptance, difficulty in recognition, reduced self-awareness. The total score correlates positively with negative affect and negatively with positive affect.

***Five Facet Mindfulness Questionnaire*** (*FFMQ* [78], translated and validated into Italian [79]). It is a 39-item questionnaire that evaluates the aptitude of the person towards mindfulness. Items are scored on a 5-point Likert-type scale ranging from 1 (never or very rarely true) to 5 (very often or always true). Five dimensions are identified: observation, description, conscious actions, non-judgmental inner experience and non-reactivity.

***Psychophysiological evaluation:*** cardiac activity (ECG), electrodermal activity (EDA), respiratory activity (RESP) and temperature (T) will be collected through the EEG telemetry and polygraphic system “Encephalan-EEGR-19/26”. A tripolar ECG sensor will be applied with proximal disposition at subscapular sites. The EDA sensor (superimposing an alternate current >50 Hz so as to not interfere with the other data acquisition, with conductance range 1–100 µS) will be placed over the index and ring finger of the non-dominant hand. The respiratory belt will be applied to the abdominal site, and the temperature skin sensor will be placed on left little finger. The sampling rate will be set to 250 Hz. The “Encephalan-EEGR-19/26”, which includes ‘ABP-26’ and ‘Poly-10′ configuration with Rehacor (version 6 Dec 2021) and Encephalan EEG ‘Elite suite’ (version A_1744/02_13, 5.4-16-2.0 Metrology module SignalTools.dll 2.0) software, produced by Medicom MTM Ltd. (Taganrog, Russia) and distributed by Gea Soluzioni s.r.l (Turin), will be adopted for the acquisition of physiological data.

At T0b, a psychophysiological profile (i.e., ‘stress test’) will be recorded according to the observations of Thieme et al. [49] on FM psychophysiological reactivity. It consists of the non-invasive detection of physiological variables (ECG, EDA, T, RESP) during the presentation of several cognitive and emotional stimuli (such as mathematical calculation when listening to background noise, viewing images with unpleasant emotional value, the request to pay attention to the tale of stories focused on pain perception). Each stress task has a duration of 4 min and is followed by a recovery period of 4 min. At the beginning of this evaluation, the recording includes a ‘baseline phase’ in which the patient looks at neutral pictures for 6 min. The entire psychophysiological assessment has an expected duration of 36 min.

The psychotherapeutic intervention will then take place and will last from **T1 to T12,** followed by a final post-treatment **(T13)** evaluation and **two follow-up sessions**, at **one** and **six months** after the conclusion of the intervention. The follow-up will be based on the same psychophysiological assessment performed at T0 (with different versions of the stress task) and the administration of the FIQ-R, BPI, DERS and PSEQ scales. At 6 months, the follow-up will be based on the FIQ-R, BPI, DERS e PSEQ scales.

At **T2, T5, T9 and T12**, continuous and synchronous recording of **physiological variables** (RESP, ECG, EDA) of both patient and therapist will be collected during the whole psychotherapy consultation and will be arranged as shown in Figure 2.

The researcher acquiring computer data will stay in an adjacent room to that of the therapist and the patient, which will be connected with a one-way mirror to allow the supervision and the correct detection of physiological parameters.

We chose T2-T5-T9-T12 for physiological recording for different reasons:T2 was chosen to allow the creation of a basic relationship between the patient and psychotherapist after the first sessions;T5 was selected because the literature in this field showed evidence for a linear increase in the synchronization of physiological parameters, particularly during the third–fourth session of psychotherapy [40,80];T9 was considered as a good intermediate point between T5 and the final session at T12;Lastly, T12 indicates the final quality of therapist–patient interaction.

The session will also be video-recorded after the patient’s approval and transcribed to permit a detailed analysis of the patient and therapist interaction.

Before starting T2-T5-T9-T12 consultations, the patient will complete the MPQ scale and will undergo a 5 min baseline recording of physiological parameters while remaining in an emotionally neutral condition.

At the end of the consultation, the visual analogue scale of empathy (VAS-E), the working alliance inventory short revised (WAI) and the perceived empathy scale (SEP) (only at T2 and T12) will be collected for both the patient and therapist. The patient will also be asked to report which moments of the consultation were felt as more empathic to better understand which moment of the consultation contributed more to empathy perception.

***Visual Analogue Scale-Empathy****(VAS-Empathy*), on a 1–10 scale. Both patient and therapist indicate the degree of empathy perceived during the whole consultation and describe the moment in which he/she felt most attuned (referring to the interview that has just ended).

***Working Alliance Inventory Short Revised*** (*WAI SR* [81]). It is a 12-item questionnaire on a 5-point Likert scale rang ranging from 1 (never) to 5 (always) that evaluates three dimensions of therapeutic alliance as evaluated both by the patient (WAI-SR-C) and the therapist (WAI-SR-T): the quality of the interpersonal bond between the therapist and client (bond), the agreement on the therapy tasks (task) and the agreement on the therapy goals (goal). ***Perceived Empathy Scale*** (*SEP* Italian version [82] of the empathic understanding EUS, subscale of the Barrett-Lennard relationship inventory [83]). It is made up of two parts: “Me-towards-the-Other” (EUS-MO, in Italian SEP-M ‘Me verso Altri’), evaluating the therapist’s perceived ability in transmitting empathy towards the other member of the dyad; “Other-to-Self” (EUS-OS, in Italian SEP-A ‘Altri verso Me’), reporting the therapy client’s perceived empathy. Each form (SEP-A and SEP-M) is composed of 16 items rated on a 6-point Likert scale (from −3 ‘strongly disagree’ to +3 ‘strongly agree’). Total scores range between −48 and +48, with higher scores indicating higher perceived empathy.

Table 1 gives a summary description of the timeframe of each evaluation.

### 2.3. INTEGRO Intervention

It is a manualized psychotherapeutic intervention based on CBT, ACT and mindfulness techniques, in line with already published protocols, such as FibroQoL [34,35,84] (FibroQoL Study Group), and FM treatment indicated by Thieme, Flor and Turk [33].

The INTEGRO intervention gives specific guidelines in order to minimize inter-therapist variability, while ensuring adequate inter-patient flexibility.

The intervention is based on 11–12 individual sessions, once a week, of one hour each and is divided into 3 macro phases (Table 2).

During the first phase, it is essential to share with the patient the rationale behind the integration of psychotherapy and pharmacotherapy (if any), the objectives of the intervention and the role of cognitive and emotional variables in the perception of pain, together with psychoeducation on chronic pain.

Contingencies between pain and contextual patient reactions together with maintenance factors of stress and pain are deeply assessed during the second phase. The protocol sessions will also include: teaching and practice of self-monitoring skills to increase pain awareness, learn relaxation techniques (i.e., guided imagery focused mainly on body temperature) to manage and reduce pain perception, cognitive restructuring, mindfulness and defusion ACT-based techniques and implementation of effective and assertive communication with doctors and family members to improve performance in the long term.

Finally, the third phase consists of a final interview that aims to consolidate all attainments achieved during the whole intervention.

### 2.4. Study Objectives and Related Endpoints

The main objective related to INTEGRO intervention is to evaluate QoL perception before pre-treatment (T0) and after post-treatment (T13) intervention, as measured by the FIQ-R scale (primary endpoint).

Other objectives (secondary endpoints) are:to assess whether INTEGRO intervention contributes to reducing the intensity of pain perception, in terms of median score difference on the BPI scale between T0 and T13;to evaluate patients’ perception of self-efficacy in managing pain by calculating median scores differences in the PSEQ scale between T0 and T13;to assess patients’ emotional regulation abilities in terms of median scores difference on the DERS scale between T0 and T13;

The SEP (SEP-A/SEP-M, degree of empathy between patient and therapist) and WAI (WAI-C/WAI-T, degree of therapeutic alliance between patient and therapist) scales as well as the physiological concordance index (derived from the measurements of EDA and ECG between patient and therapist in T2, T5, T9 and T12) will be considered as mediating variables. For this reason, we will analyze the association between SEP, WAI and physiological concordance index and also their contribution to explain the change in QoL perception, BPI, PSEQ and DERS scales between T0 and T13.

Traumatic experiences, attachment style, mindfulness attitude and psychopathology will be considered as covariates.

### 2.5. Sample Size Calculation

We assume that we have to recruit 45 patients to have a final sample of 38. Sample size was calculated according to the recruitment capacity of the Pain Therapy Unit and on dropout percentages (around 20%) reported in the literature [35,85,86].

### 2.6. Physiological Signals Processing

ECG signals will be processed to derive the heart rate variability (HRV) series, which represents the variability of the time distance between two consecutive R peaks (RR interval). ECG data collection, preparation (including, e.g., filtering) and HRV calculation will be performed according the current guidelines [87]. Eventual physiological and algorithmic artifacts will be removed, and several features defined in the time and frequency domains will be obtained from the artifact-free HRV series according to the current guidelines [88]. Time domain features will include: mean of the RR intervals [msec], SDNN = standard deviation of the RR intervals with respect to the mean [msec], RMSSD = square root of the mean squared differences between adjacent RR intervals. For the frequency domain features, a spectral analysis will be performed in accordance with the Welch estimation method, as well as through the implementation of a parametric method based on the AR model (autoregressive). Such an analysis will allow the calculation of the power in the HF (high-frequency) band defined between 0.15 and 0.4 Hz. Note that SDNN, RMSSD and HF power may be considered as biomarkers correlated with vagal activity (assuming the respiratory frequency is within the HF band) [88], and HRV indices may serve as bio-markers of top-down self-regulation [48].

The respiratory signal will be pre-processed using bandpass filters between 0.01 Hz and 2 Hz, and the following features will then be derived: respiratory rate (number of breaths within a given time window, usually lasting one minute), mean variation of the signal, expressed as the average of the first derivative, signal strength. The application of multivariate measures for the estimation of cardiorespiratory coupling by means of phase lock and synchrogram measurements will eventually be evaluated as well.

The electrodermal response will be filtered to remove frequency components higher than 2 Hz and will then be decomposed into two signals concerning a basal level (SCL) and a phasic level (SCR). From each of these signals, descriptive measures such as mean and variance will be derived.

Time-resolved information about skin temperature will be considered through mean and variance over time.

The aforementioned estimates of the autonomic activity derived from the physiological signals (ECG, respiratory signal, electrodermal response) will be derived from both the therapist and patient, and the associated level of agreement will be calculated. In particular, a linear correlation analysis will be performed on such paired data considering different experimental phases. Furthermore, considering HRV series recorded in the patient and in the therapist in sync, synchronization indices such as coherence and phase synchronization indices (e.g., so-called phase-locking values) will eventually be calculated.

### 2.7. Statistical Analysis

To check statistical assumptions before running any analysis, preliminary testing for normality will be assessed group-wise for each psychometric score and for each feature from physiological signals. Parametric or non-parametric tests will then be applied accordingly.

For the analysis of all objectives of the study (main and secondary), a paired t-test or a paired Wilcoxon test will be performed.

To deal with the multiple measures collected by the mediating variables (SEP, WAI, psychophysiological index), generalized mixed models for repeated measures on paired data will be performed.

To evaluate the association between mediating variables, the Spearman or Pearson coefficient will be investigated.

Finally, appropriate regression models, considering covariates, will be built to evaluate the role of mediating variables on the main and secondary objectives.

For all analyses, a 5% significance level will be considered, and a statistical correction for multiple comparison will eventually be applied according to the experimental hypotheses. Appropriate 95% confidence levels or interquartile ranges will be produced.

## 3. Discussion

As described in the introductive section, FM patients show a constellation of both somatic (diffuse muscle pain, sleep disorders, irritable bowel, intimate burning, chronic fatigue) and psychological symptoms (the so-called fibro fog, that is, difficulty in concentrating and performing simple mental processing, high levels of anxiety, depression, high focus on painful stimuli, catastrophizing and greater sensitivity in the dynamics of interpersonal identification and social comparison) [8,9]. Among psychological symptoms, Terol et al. [89] reported that FM patients tend to “*compare themselves with others on “similar health problems” and on contents such as “symptoms”, showing a psychological profile different from findings in other chronic patients (rheumatoid arthritis or cancer patients)*”and may be distinguished into those who have a “maladaptive” profile, characterized by higher levels of pain perception, anxiety and depression and more frequent “unfavorable” social comparison strategies and those who show a more “adaptive” profile (moderate levels of pain perception, lower level of anxiety and depression and more frequent “favorable” social comparison strategies). It is not yet entirely clear which is the role of these psychological factors in the onset and maintenance of organic pathology.

The literature on FM psychological interventions tends to support multicomponent or multidisciplinary interventions, which include the management of both psychological and physical symptoms by combining CBT interventions with physical exercise, physical therapy or drug therapy, demonstrating that they are superior to a single pharmacological treatment [13,17]. Nevertheless, most of the literature reviews tend to exclude multicomponent interventions, being difficult to clearly distinguish the effect of each type of technique [31]. Moreover, the literature tends to report mainly the results of group-specific therapies [21,22,28,32], compared to individual ones, which are more variable in relation to patient characteristics, require greater economic resources, time and staff availability. Therefore, on one side, evidence-based approaches and recent guidelines [13,14] suggest adopting interventions tailored to the needs of each FM patient as much as possible, including a variety of approaches (i.e., acupuncture, biofeedback, manual therapy, CBT, exercise, hydrotherapy, hypno-therapy, massage, meditative movement, mindfulness/mind–body therapy, multimodal therapy), and on the other side, general reviews report more specific interventions, mainly based on group approaches, making it difficult to understand the role of individual characteristics in therapeutic outcomes.

Based on these observations, we chose to conduct a pilot study based on an individualized integrated and multicomponent evidence-based intervention [33,34,35,84] in which we combine CBT with specific techniques of self-monitoring skills to increase pain awareness, relaxation techniques to manage and reduce pain perception, mindfulness ACT-based techniques and assertive communication with doctors and family members to improve performance in the long term. Therefore, this intervention aims to promote in each FM patient a better ability to process painful experiences, including a greater awareness of the interaction between pain-related somatosensory, cognitive, affective and behavioural factors.

In addition, the effectiveness of the intervention on QoL, pain-managing self-efficacy, emotion-regulation abilities and pain intensity perception will be related to therapeutic relationship variables (i.e., therapeutic alliance, physiological attunement, perceived empathy), considering individual characteristics as covariates (i.e., attachment dimensions, lifetime traumatic experiences, abilities in emotion regulation and mindful thinking). This is because the recent literature has reported some evidence on the role of patients’ attachment dimensions on the quality of therapeutic alliance [36,90,91] suggesting that secure patients tend to have more collaborative and change-oriented attitudes, contributing to more favorable therapeutic outcomes. Nevertheless, most of the studies reported in the literature are based on conscious reports (i.e., working alliance inventory, adult attachment interview) and very few consider both conscious and less conscious (i.e., psychophysiological attunement [40]) processes. According to Smith, Msetfi, and Golding [47], “*it is important to be mindful that a measure asking clients to rate their attachment in general adult relationships may not find results exactly equivalent to a measure asking clients to rate their relationship with their therapist*”. This can be particularly relevant when accounting for difficult therapeutic relationships, where poor alliance and insecure attachment may be present. There is some evidence that the more the patient has secure attachment-based relationships, the better the quality of the alliance with her therapist will be, and on the other side, there is less evidence for patients with an insecure attachment.

In our opinion, in order to untangle this complex interaction, the process should be analyzed with different coding and interpretation keys, which include both conscious (verbal or written reports) and less consciously controllable aspects (such as physiological reactivity, i.e., concordance indices of skin conductance and heart rate variability [92]), together with observational coding systems of the interaction between the patient and the therapist, as additional indicators of the quality of attachment and reciprocal relationship (such as, for instance, the patient attachment coding system—PACS [93]—and the therapist attunement scales—TASC [94]). As suggested by Nyman-Salonen et al. [95], embodied aspects are one of the “common factors shared among psychotherapy approaches that account for the effectiveness of the treatment” [95,96,97,98]. This because part of the change during the psychotherapy process occurs through the embodied connection between the therapist and client, revealing important bonding aspects which cannot be symbolized verbally, particularly when considering patients with somatising disorders. Other individual variables that are clinically relevant in explaining the etiopathogenesis and the exacerbation of chronic pain in FM patients are lifetime traumatic experiences, emotion regulation and mindful thinking inabilities.

Lifetime traumatic experiences and chronic stress may contribute to the onset and maintenance of chronic pain [99,100] because trauma can lead to an interruption of the normal integration of mental functions, such as the sense of self identity, body representation, emotional perception and control [101]. Romeo et al. [99], for instance, showed that patients with FM obtained significantly higher scores in the traumatic experiences checklist (TEC [75])—a self-report scale that measures exposure to potentially traumatic events, including actual or threatened bodily harm to self or others, and emotional neglect and abuse, as well as physical and sexual abuse—compared with a pain-free healthy group. Specifically, patients with FM reported more frequent traumatic events during childhood, greater impact of trauma and cumulative trauma, with a tendency to show worse forms of somatoform dissociation in those who had more severe FM symptoms.

Trucharte et al. [102] showed that FM patients have difficulties in adjusting their emotional states, particularly in relation to the very understanding of emotions, and present emotional interference in interpreting inner states, which contributes to a greater impact of pain in daily activities, together with an increased disability burden and the feeling of being overwhelmed by intensity and persistence. Some authors related the affective-sensory dimension of pain to the difficulty to name emotions (alexithymia) [103]. Alexithymia seems to interfere with an adequate regulation of emotions, increasing negative affective states, which, in turn, shapes the affective dimension of pain experience [104]. Alexithimic aspects defined as “emotional awareness” are analyzed within the DERS scale in our study and might be an object of specific interest.

Finally, mindfulness abilities are strictly related to emotion regulation. Thus, a mindfulness-based intervention, by adopting a nonreactive and nonjudgmental focus, can enhance cognitive flexibility and the ability to regulate emotional reactivity, contributing to reduce pain and to improve overall health and functioning [35,105]. However, as observed by Adler-Neal and Zeidan [105], not all FM patients have enough cognitive resources to manage mindfulness-based techniques well, which might be perceived by some patients as an overly demanding cognitive task requiring too much sustained attention. Therefore, following these authors’ suggestion, the use of less cognitively engaging non-pharmacologic approaches (such as relaxation, biofeedback, acupuncture and yoga) may prove to be more helpful in reducing stress, anxiety and pain perception for some patients. For this reason, in our study, once specific mindfulness abilities (i.e., FFMQ evaluation before the intervention [78]) are evaluated, we combine mindful-based with cognitively less demanding relaxation techniques as a strategy to better shape the intervention to each patient’s characteristics.

### Limitations

Although physiological measures may reflect autonomic phenomena which are not consciously controllable, revealing unaware aspects of the quality of the therapeutic relationship, they do not allow univocal meanings and reflect a non-specific physiological activation, which needs careful interpretation [39,95,106].

Moreover, the equipment for detecting psychophysiological signals both in the patient and the therapist, albeit managed in such a way to be as noninvasive as possible, may contribute to altering the genuineness of the interaction.

Furthermore, the need to set eligibility criteria that consider physiological measure constrains may be a source of bias when selecting patients, reducing sample general representativeness with respect to the FM population. This primarily relates to the most seriously ill patients, who are excluded from psychophysiological detection because of their organic conditions or painkiller drug use.

Finally, the preliminary structure of the research and the complexity of the intervention do not allow the study to be carried out in a large sample nor to implement a clinical trial including a control group. Maybe this will be possible after preliminary results coming from the pilot study.

## 4. Conclusions

To conclude, this protocol will provide a promising individualized and patient-tailored treatment, relating the results to the analysis of the role of the therapeutic relationship together with patient characteristics.

The preliminary results of this study may provide a basis for further research in this field, obtaining important results for FM patients, therapists and clinicians involved in chronic pain management.

## Figures and Tables

**Figure 1 ijerph-20-03973-f001:**
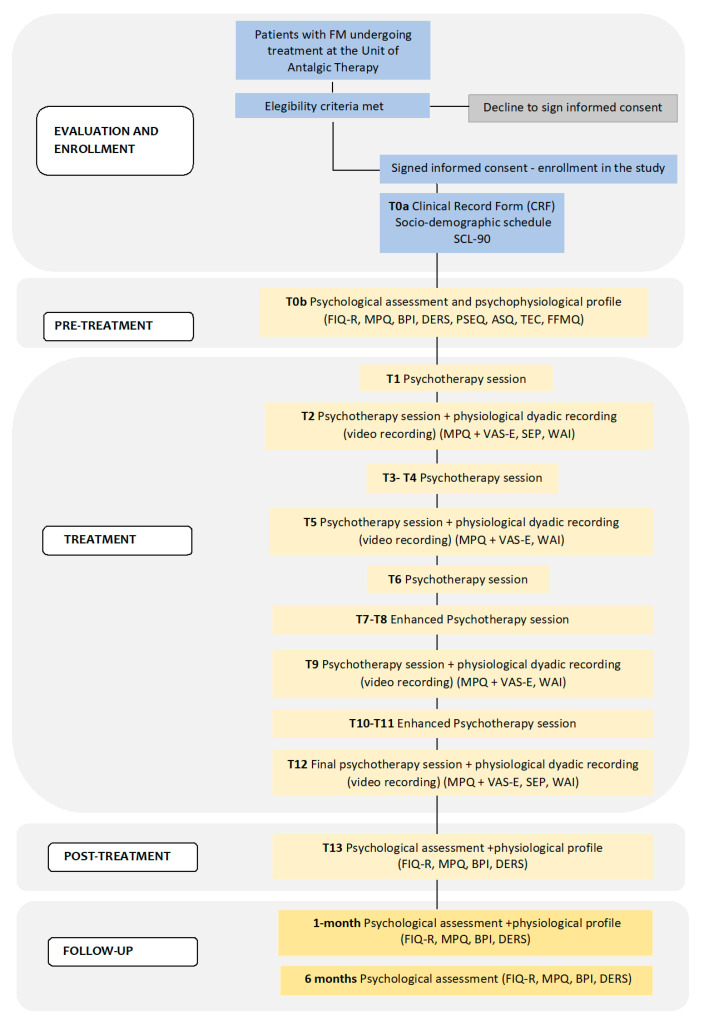
Flowchart of INTEGRO study.

**Figure 2 ijerph-20-03973-f002:**
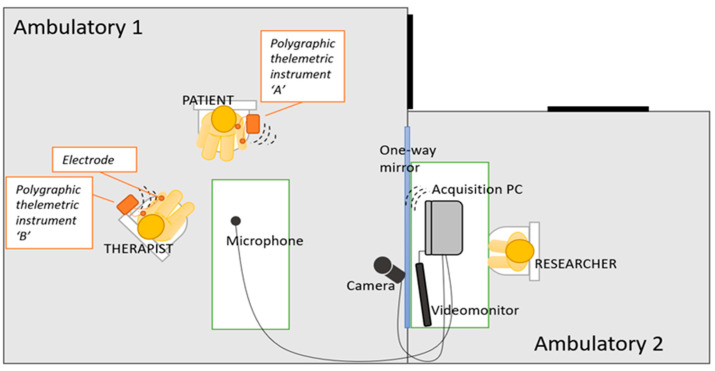
Organization of space during psychophysiological recording sessions.

**Table 1 ijerph-20-03973-t001:** Evaluation timeframe.

	Study Period													
	First Visit	Pre-Treatment	Treatment									Post-Treatment	Follow-Up	
Evaluation and Tools	T0a	T0b	T1	T2	T3 T4	T5	T6	T7 T8	T9	T10 T11	T12	T13	1 Month	6 Months
*Eligibility screening*	**●**													
*Informed consent*	**●**													
*Clinical record form and socio-demographic*	**●**													
*SCL-90*	**●**													
*Psychophysiological profile*		**●**										**●**	**●**	
*Psychophysiological recording*				**●**		**●**			**●**		**●**			
*Videomonitoring **				**●**		**●**			**●**		**●**			
*FIQ-R*		**●**										**●**	**●**	**●**
*DERS*		**●**										**●**	**●**	**●**
*BPI*		**●**										**●**	**●**	**●**
*MPQ*		**●**		**●**		**●**			**●**		**●**			
*PSEQ*		**●**										**●**	**●**	**●**
*ASQ*		**●**												
*TEC*		**●**												
*FFMQ*		**●**												
*VAS-E ***				**●**		**●**			**●**		**●**			
*SEP-A and SEP-M ***				**●**							**●**			
*WAI-C and WAI-T ***				**●**		**●**			**●**		**●**			

* only by specific patient consent submission, ** involve patient and therapist, ● is assessed.

**Table 2 ijerph-20-03973-t002:** INTEGRO intervention.

	Treatment Details
**PHASE 1**	**ENGAGEMENT, MOTIVATION-SHARING, PSYCHOEDUCATION, COGNITIVE RESTRUCTURING**
T1	*Definition of the problem reported by the patient, the history of pain, personal and environmental factors that contribute to the maintenance, worsening and improvement of pain. Meaning attributed to the disease and to the pain symptom.*
T2 *	*Perception of pain through the McGill Pain Questionnaire. Psychoeducation. Diary of the pain.*
T3-T4	*Diary analysis of pain, pain-related behaviors and thoughts. Psychoeducation.* *Coping strategies and maintenance cycles in pain, analysis of ‘pain behaviors’. Sharing of situations in which the patient perceives a poor management of painful symptoms with identification of situational antecedents (A), thoughts (B), behavior and emotions (C).*
T5 *	*Perception of pain through the McGill Pain Questionnaire. Influence of chronic pain on interpersonal dynamics. Analysis of situations in which the patient perceives a poor management of painful symptoms through the ABC technique. Cognitive restructuring.*
T6	*The role of control in chronic pain, patient attitudes and beliefs. Analysis of ‘pain behaviors’. Pain diary with distinction of the physical sensory component from the emotional component.*
**PHASE 2**	**COGNITIVE RESTRUCTURING, REDUCTION IN AVOIDANCES AND PROMOTION OF ALTERNATIVE BEHAVIORAL STRATEGIES, INTRODUCTION OF EXPERIENTIAL AWARENESS TECHNIQUES AND THOUGHT–ACTION DEFUSION**
T7-T8	*Pain diary with distinction of the physical sensory component from the emotional component. The role of awareness of enteroceptive stimuli in chronic pain. Experiential techniques of body awareness, breath and body temperature.*
T9 *	*Perception of pain through the McGill Pain Questionnaire and the diary. Goals and values that guide behavior.*
T10-T11	*Analysis of ‘pain behaviors’. Cognitive defusion. Experiential techniques of body awareness, exploration of pain, acceptance and acceptance of pain.*
**PHASE 3**	**CONCLUSION**
T12 *	*Perception of pain through the McGill Pain Questionnaire. Conclusion of the process and the asset acquired.*

* In these interviews, the synchronous revelation of the physiological parameters of the patient and therapist is envisaged, and the video recording of the interview is only upon specific consent.

## Data Availability

Data will be available on motivated request sent to Lidia Del Piccolo.

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
