# Peer review of "“INTEGRO INTEGRated Psychotherapeutic InterventiOn” on the Management of Chronic Pain in Patients with Fibromyalgia: The Role of the Therapeutic Relationship"

_ijerph, 2023, doi:10.3390/ijerph20053973_

Round 1

Reviewer 1 Report

Authors treat a very important and controversial topic: the management of chronic pain in patients with Fibromyalgia. But they choose an important point of view: the role of the therapeutic relationship.

 It is very important topic, authors should consider the role of mood and the pharmacological treatment (10.1017/S1041610217001715 and 10.1016/j.jamda.2019.01.128) on this patients, especially in older adults.

To reduce pharmacological treatment in fragile patients is very important, and for FM patients is possible implementing the role of psychotherapy. It is complementary to drugs and physical activity (10.1016/j.archger.2020.104109).

Author Response

We thank the reviewer for the two suggested references. We added both [ref. 15-16] in the paragraph at lines 58-66.

In any case we will include 18-65 years old patients, not older as indicated at line 170.

Reviewer 2 Report

This is a study protocol for an integrated psychotherapeutic intervention for patients with Fibromyalgia (FM), whose aim is foster a better management of chronic pain among patients with Fibromyalgia. This is an interesting study protocol for an important yet underinvestigated area of research in health\clinical psychology. I commend the authors for their well-conceived study protocol. However, I've some concerns, which are detailed below:

Abstract

I noticed that the authors use the term "attachment styles". To my understanding, self-report measures of attachment examine this construct only from a dimensional point of view (i.e., how anxiously-, avoidantly- o securely-attached an individual is, within a dimension that goes from low to high), so I would refer to attachment dimensions -rather than styles- throughout the manuscript.

Introduction

1) Section 1.1 use the term "chronic pain" as a synonim of Fibromyalgia. I'm not an expert in this area, but according to ICD-11 isn't Fibromyalgia one of the several possible chronic pain conditions? thus, I think it would be clearer for the reader if authors stick to the same term throughout the manuscript, or at very least clearly define that they will refer to "chronic pain" as a synonim of Fybromialgia at the beginning of the introduction.

2) sentence 56-57 is not clear to me. In the previous paragraph (line 48-49) you mentioned that the etiology of FM is not entirely clear, but then (in lines 56-57) you write that psychotherapy is useful especially for "for the treatment of chronic pain whose etiology is strongly related to psychological factors". So, are there sub-types of patients, some with a more psychologically-driven etiology, and some with a more-biological\unexplained etiology?

3) line 68, "The primary outcomes highlighted are": highlighted where? in the previously-mentioned reviews?

4) Authors should explain in a better way why it is important to study psychophysiological responses among patients with FM, and provide a better rationale for the use of EEG (if these data are going to be collected, of couse), HRV and EDA signals in psychotherapy research. Besides the importance of within-session synchrony between a patient and his\her therapist, are there studies that examine pre-to-post changes in these signals among patients with chronic pain\FM after a psychological treatment? Further, the Introduction is still missing a point on how autonomic activity (especially vagal one) can be viewed as a biomarker of top-down regulatory competencies (e.g., Thayer et al., 2012; Ruiz Vargas et al., 2016; Holzman & Bridgett, 2017). This, in my opinion, would strengthen the study design, providing a stronger rationale on why to examine these physiological indices within the study (and also provide a rationale on the supposed changes in physiological indices from pre- to post-treatment).

Methods

1) In section 2.1 (including the title) I would refer to the "Study Population" (i.e., patients with FM, how do you plan to recruit them, the inclusion\exclusion criteria etc.), rather than referring to the "sample". You still do not have a sample, but rather -being this a study protocol- you plan to recruit one from a population, which is going to be described in this section.

2) line 165-166: "Only if a psychiatric history emerges; further psychopathological investigation will be conducted using the Structured Clinical Interview - Clinician Version - SCID-CV [51], Italian version [52]". If I understand correctly, patients will be interviewed with SCID-CV only if a history of psychiatric treatments emerge. Isn't this a bit limiting? what if a patient has never been hospitalized for a previous mental health problem, or what if he\she does not self-disclose (nor is aware of) a diagnosis? I understand this is also a matter of time and of resources invested in the project, but maybe another criteria to administer the SCID-CV could be -for example- if the patients' scale scores at the SCL90 are higher than a certain cut-off (or something similar), or to simply screen everyone?

3) why are there two separate pre-treatment evaluations, in two separate units? I would explain this in a better way within the manuscript

4) The methods section sometimes is difficult to follow, because it refers to things that are detailed only later, and some information are missing. Further, in line 252-254 authors mention a stress task that was not introduced in the protocol before (nor detailed later). I think the manuscript would improve providing a short overview of the entire protocol at the beginning of the Materials and Methods section (or maybe at the end of the introduction?) and then detail the different study sections in separate paragraphs (e.g., study objectives/hypotheses; study design; study population & recruitment procedure; treatment intervention; outcomes [e.g., the self-report measures; arranged in primary and secondary outcomes]; sensors; physiological signals processing; psychophysiological experiments; sample size; statistical analysis).

5) Given that study participants seem to perform also a stress task, authors should better address this in the Introduction (which -in the current version of the manuscript- mostly focus on "Interpersonal psychophysiology and therapeutic alliance")

6) how will the ECG\PPG and EDA data be collected? through the ABP-26? Will EEG data be collected too? I would suggest authors to provide also more information about the sensors: What's the sampling frequency of the devices? Cardiac activity will be monitored through an ECG or a PPG? Further, the section "Psychophysiological evaluation" does not detail the experimental procedures, but rather describe a sensor and provide limited information on the experiments\experimental settings. For example, how will the recordings be performed? will the authors acquire the physiological signals during an entire therapy session? which data will be used (e.g., they will examine synchronization during the entire session; or focus only on the first 5 minutes of the session, the 5 in the middle and the final 5 minutes)? what about the stress task(s) and its subsequent experimental procedure? will the data from the laboratory experimental tasks be acquired during the same day of the therapy session, or on a different day? Further, a "Psychophysiological profile" is mentioned in Table 1, but this is not addressed in this section. What do the authors mean? will they colled (as I suppose) resting-state physiological (HRV, EDA) data among patients, while sitting still and relaxing\performing a simple task with a low cognitive load for some time? Finally, how will the videomonitoring\videorecordings data be used? I suppose this is related to the investigation of synchronicity among patient and therapist during the three therapeutic sessions, but this has not been mentioned in the methods section of the manuscript yet.

7) Outcomes: to my understanding, most of the information provided in this section are hypotheses to be tested\study objectives, rather than primary or secondary outcomes. A primary outcome would be -for example- QoL, while secondary outcomes are all other dimensions investigated by the remaining measures (e.g., pain perception, perception of self-efficacy, emotion dysregulation; and probably the physiological indices too); on the other hand, a study hypothesis may be (i) that patients will experience longitudinal increases in QoL from pre- to follow-up after the therapeutic intervention, or that (ii) some variables (e.g. attachment insecurity) could mediate\moderate these changes. What if this section, opportunely revised and trimmed down, is rephrased as "study objectives" and moved soon after the first overview of the study protocol?

Statistical Analysis:

1) Overall, I think this section should explain which analyses authors will adopt to test the hypotheses mentioned in the "study objectives" (or the study hypotheses), providing a-priori, pre-planned analyses for the main report(s) that may come out from this study protocol.

2) line 363-364, "The Gaussianity of the related random variable will be assessed group-wise for each psychometric score". This sentence is difficult to understand. Are authors implying they will check statistical assumptions before running any analysis?

3) What about dropouts? will an intent-to-treat approach be adopted, or dropouts will be excluded from analyses?

4) I strongly advise authors to adopt mixed models to examine longitudinal changes in outcomes, because ANOVAs delete missing cases listwise and this would lower the overall sample size.

Physiological signals processing:

1) I would move this section before the statistical analysis one (which usually conclude the methods section of a manuscript). As for the HRV, I would only mention those indices which are clearly related to the vagal activity over heart (HF & RMSSD). Finally, will authors collect data, clean the signals, report data etc. all according to guidelines (e.g., in the area of HRV, Quintana, Alvares, & Heathers, 2016)? a statement like this for all physiological signals would strengthen the methodological quality of this section.

Discussion

1) fibro-fog is mentioned in the discussion, but this term has not been introduced in the introduction.

2) line 497-499, authors mention alexithymia, which is not going to be investigated in this study. I think this part could be safely removed from the manuscript

3) overall, I think this entire section could be trimmed down so to have more space in the methods section to detail in a better way the entire study protocol.

Thanks for the opportunity to review this,

Author Response

We thank reviewer 2 for the detailed revision and suggestions given.

For the responses see attached file.

Reviewer 3 Report

This is an important study on fibromyalgia. In page 2, line 50, please add: Somatosensory amplification may be associated with fibromyalgia. Please add the reference: Ciaramella A, Silvestri S, Pozzolini V, Federici M, Carli G. A retrospective observational study comparing somatosensory amplification in fibromyalgia, chronic pain, psychiatric disorders and healthy subjects. Scand J Pain. 2020 Nov 2;21(2):317-329. doi: 10.1515/sjpain-2020-0103. PMID: 34387956.

Author Response

We thank the reviewer for the positive comment.

We added the reference [12] as requested at line 57.